

# Comparative studies of two AA10 family lytic polysaccharide monooxygenases from *Bacillus thuringiensis*

Huiyan Zhang[*], Haichuan Zhou[*], Yong Zhao, Tang Li and Heng Yin

Biotechnology Department, Dalian Institute of Chemical Physics, Chinese Academy of Sciences, Dalian, China

[*] These authors contributed equally to this work.

## ABSTRACT

*Bacillus thuringiensis*, known to be one of the most important biocontrol microorganisms, contains three AA10 family lytic polysaccharide monooxygenases (LPMOs) in its genome. In previous reports, two of them, *Bt*LPMO10A and *Bt*LPMO10B, have been preliminarily characterized. However, some important biochemical features and substrate preference, as well as their potential applications in chitin degradation, still deserve further investigation. Results from present study showed that both *Bt*LPMO10A and *Bt*LPMO10B exhibit similar catalytic domains as well as highly conserved substrate-binding planes. However, unlike *Bt*LPMO10A, which has comparable binding ability to both crystalline and amorphous form of chitins, *Bt*LPMO10B exhibited much stronger binding ability to colloidal chitin, which mainly attribute to its carbohydrate-binding module-5 (CBM5). Interestingly, the relative high binding ability of *Bt*LPMO10B to colloidal chitin does not lead to high catalytic activity of the enzyme. In contrast, the enzyme exhibited higher activity on β-chitin. Further experiments showed that the binding of *Bt*LPMO10B to colloidal chitin was mainly non-productive, indicating a complicated role for CBM5 in LPMO activity. Furthermore, synergistic experiments demonstrated that both LPMOs boosted the activity of the chitinase, and the higher efficiency of *Bt*LPMO10A can be overridden by *Bt*LPMO10B.

Corresponding author
Heng Yin, yinheng@dicp.ac.cn

## INTRODUCTION

Chitin can be considered as the second most abundant biopolymer on Earth. It consists of β-1,4-linked N-acetylglucosamine and is widely distributed in the exoskeleton of crustaceans and in the cell walls of insects and fungi (*Tharanathan & Kittur, 2003*). Chitin can be classified into α (anti-parallel chains) and β (parallel chains) crystalline forms. Chitin is responsible for providing important characteristics such as rigidity and strength to the cell wall, and it also adds to the defense of the cell against pathogens and various predators (*Beckerman et al., 2013*; *Bowman & Free, 2006*; *Brunner et al., 2009*; *Vincent & Wegst, 2004*). Thus, the destruction of the crystalline structure of chitin in insects or fungi with the application of chitin-degrading microorganisms has been thought to be a broad-spectrum biocontrol strategy in agriculture (*Le & Yang, 2019*). Besides, the conversion

of chitin waste, such as shrimp and crab shells, into chitooligosaccharides contributes to increased nutritional benefits in the food industry (*Le & Yang, 2019*).

In nature, chitin-degrading organisms have evolved a series of enzymes that are involved in the synergetic depolymerization of chitin, including glycoside hydrolases (GHs) and lytic polysaccharide monooxygenases (LPMOs) (*Vaaje-Kolstad et al., 2013*; *Vaaje-Kolstad et al., 2010*). GHs typically have their activity targeted on the amorphous region (non-processive enzymes) or the end of the chitin chains (processive enzymes) and cleave the glycosidic bond using a hydrolytic mechanism (*Vaaje-Kolstad et al., 2013*). On the other hand, LPMOs are recently discovered as copper-dependent metallo-enzymes which can oxidatively destroy the crystalline region of the recalcitrant polysaccharides and are also responsible for boosting the efficiency of GHs (*Merino & Cherry, 2007*; *Vaaje-Kolstad et al., 2010*). Due to their promising application in biomass bioconversion and biorefinery, LPMOs have been given special attention (*Johansen Katja, 2016*; *Martínez et al., 2017*; *Monclaro & Filho, 2017*). In addition, some researchers have proposed that LPMOs may also be associated with the virulence of the pathogens (*Agostoni, Hangasky & Marletta, 2017*; *Paspaliari et al., 2015*; *Sabbadin et al., 2021*; *Wong et al., 2012*).

*Bacillus thuringiensis* is one of the most important biocontrol microorganisms, which has been used in agriculture as a biopesticide for a long time to control various invertebrate species (*Melo, Soccol & Soccol, 2014*). Besides the cytotoxicity of organic insecticides, *B. thuringiensis* secretes diverse chitin-degrading enzymes, including LPMOs, which could affect insect growth, and ultimately lead to death of insects (*Veliz, Martínez-Hidalgo & Hirsch, 2017*; *Kramer & Muthukrishnan, 1997*; *Melo, Soccol & Soccol, 2014*). In the genome of *B. thuringiensis*, three AA10 family LPMOs encoding genes can be discovered, among which two had already been characterized (*Manjeet et al., 2019*; *Zhang et al., 2015*). *Bt*LPMO10A is a single-modular LPMO which acts on both crystalline chitin ($\alpha$- and $\beta$-chitin) and colloidal chitin, and generates products with different patterns (*Zhang et al., 2015*). In contrast, *Bt*LPMO10B (reported by *Manjeet et al. (2019)* and denoted as *Bt*LPMO10A-FL) is a multi-modular LPMO and the roles of individual domains in substrate (crystalline chitin) binding have been characterized. However, biochemical features such as the effect of substrate binding on $H_2O_2$ generation, as well as their synergistic activity with chitinase in chitin degradation still worth further investigation. In this study, the biochemical features and substrate preference of *Bt*LPMO10A and *Bt*LPMO10B were compared with the aim of identifying and understanding their functions in chitin degradation. The obtained results showed the significantly different substrate preferences of the two enzymes although they shared highly conserved substrate binding surfaces in their catalytic domain. The C-terminal domains of *Bt*LPMO10B do enhance the substrate binding ability of the enzyme, especially on colloidal chitin, whereas it has little effect on the activity of *Bt*LPMO10B. Synergetic assays indicated that the efficiency of chitinase can be significantly improved by both the *Bt*LPMO10A and the *Bt*LPMO10B, whereas the higher effect of the *Bt*LPMO10A can be attenuated by *Bt*LPMO10B.

## MATERIAL AND METHODS

### Sequence and structure analysis

The sequences of *Bt*LPMO10A (GenBank ID: AJP62637) and *Bt*LPMO10B (GenBank ID: ALP73598) can be accessed in the Genbank database and the crystal structure of *Bt*LPMO10A was obtained from the Protein Data Bank database with accession code 5WSZ (*Zhang et al., 2020*). The three-dimensional structure of the catalytic domain of *Bt*LPMO10B (*Bt*LPMO10B-CD) was generated by homology modeling using Modeller 9.19 (*Webb & Sali, 2016*) with the crystal structure of *Ba*LPMO10A from *Bacillus amyloliquefaciens* (PDB ID: 2YOX) (*Hemsworth et al., 2013*) as the template since they share the highest sequence identity (66%) (*Zhang & Madden, 1997*). After been further validated by DOPE score, a structure-based sequence alignment of *Bt*LPMO10A and *Bt*LPMO10B was conducted using Mega 7.0 (*Kumar, Stecher & Tamura, 2016*) and ESPript 3.0 (*Robert & Gouet, 2014*).

### Cloning of *Bt*LPMO10B and its catalytic domain *Bt*LPMO10B-CD

We produced three recombinant LPMOs from *B. thuringiensis kurstaki* ACCC10066 in *E. coli* BL21 (DE3), including the previously reported *Bt*LPMO10A which stored in the lab. The gene encoding *Bt*LPMO10B was amplified from the genomic DNA of *B. thuringiensis kurstaki* ACCC10066 using a forward primer F1: 5′-GGAATTCCATATGCACGGTTTTGTTGAAAAGCCCGGTA-3′ encoding a restriction site for *NdeI* and a reverse primer R1: 5′-CCGCTCGAGCACTGTTTTCCATAATGATAATGCA-3′ with a restriction site for *XhoI*. The amplified gene was then subcloned into the pET23b vector through double digestion with the two restriction enzymes. The catalytic domain of *Bt*LPMO10B (*Bt*LPMO10B-CD) was synthesized and cloned into the same vector by the Taihe Biotechnology Co., Ltd (Beijing, China). After verification by sequencing, three recombinant plasmids were transformed into *Escherichia coli* BL21 (DE3) competent cells, respectively, for protein expression.

### Protein expression and purification

The recombinant *E. coli* BL21 (DE3) cells were cultivated in 1 L Luria-Bertani (LB) medium at 37 °C with constant shaking at the speed of 180 rpm. When the $OD_{600}$ of the culture reached 0.6, a final concentration of 0.05 mM IPTG and 0.2 mM $CuSO_4$ were added and the cultivation was continued for an additional 4 h at 30 °C. Afterword, the cells were harvest by centrifugation at 4 °C for 10 min with the speed of 8,000 × g, and then resuspended in 100 mL of hypertonic solution containing 100 mM Tris–HCl pH 8.0, 20% sucrose and 0.5 mM EDTA. This step was performed two times. Finally, the precipitated cells obtained by centrifugation were resuspended in 100 mL hypotonic solution (1 mM $MgCl_2$) and incubated on ice for 10 min. After 10 min of centrifugation at 8,000 × g, the supernatant was collected for further purification.

For the purification of *Bt*LPMO10A, a chitin beads affinity chromatography method was performed as described previously (*Zhang et al., 2015*). For the *Bt*LPMO10B, a similar method was adopted with some modifications. The loading buffer was changed to 20 mM Tris–HCl (pH 8.0) and 0.15M $(NH_4)_2SO_4$, and the protein was eluted by 20 mM

acetic acid. For the purification of $Bt$LPMO10B-CD, an ion exchange chromatography with HiTrap Q column (GE Healthcare, USA) was performed. The protein solution was loaded onto the column equilibrated with 20 mM Tris–HCl buffer (pH 7.5) and eluted with a linear salt gradient using 1 M NaCl (pH 7.5). The obtained fractions were pooled and concentrated using the Amicon 8400 stirred cell (Millipore, Burlington, MA, USA) installed with a 3kDa cut-off membrane. Samples purity was analyzed by SDS-PAGE and the protein concentrations were measured by Bradford, using bovine serum albumin as a standard.

## Substrate binding assays

The reactions were conducted in 20 mM Tris–HCl (pH 8.0) buffer containing 1 $\mu$M enzyme and 5 mg mL$^{-1}$ $\alpha$-chitin, $\beta$-chitin and colloidal chitin, respectively, prepared according to the procedure described previously (*Zhang et al., 2015*). The mixture was incubated 6 h at 25 °C with constant shaking at 800rpm using Thermo block (Eppendorf, Hamburg, Germany). After been separated from the mixture by filtration through a 0.22 $\mu$m membrane, the concentrations of the free proteins measured using the Quick Start$^{TM}$ Bradford assay (Bio-Rad, Hercules, CA, USA). The mixtures without substrate were treated in the same way and used as the basis for calculating the percentage of free and bound protein.

## H$_2$O$_2$ generation assays

The reactions were conducted in 20 mM Tris–HCl (pH 8.0) buffer containing 1 $\mu$M enzyme, 1 mM ascorbic acid and 5 mg mL$^{-1}$ $\alpha$-chitin, $\beta$-chitin and colloidal chitin, respectively. The mixture was incubated 2 h at 30 °C with constant shaking at 800 rpm using Thermo block (Eppendorf, USA). After been separated from the reaction mixture by filtration through a 0.22 $\mu$m membrane, the concentrations of H$_2$O$_2$ in the supernatant were measured using the Fluorimetric Hydrogen Peroxide Assay Kit (Sigma, St. Louis, MO, USA). The reactions without substrate were set as the control.

## Enzymatic reactions

Enzymatic reaction was performed in a 500 $\mu$L reaction mixture containing 5 mg mL$^{-1}$ substrate, 20 mM Tris–HCl (pH 8.0), 1 $\mu$M enzyme, and 1 mM ascorbic acid. For the reaction using both $Bt$LPMO10A and $Bt$LPMO10B, 0.5 $\mu$M $Bt$LPMO10A and $Bt$LPMO10B was added. The reaction was last for 16 h at 30 °C with constant shaking at the speed of 800 rpm. After been separated from the reaction mixture by filtration through a 0.22 $\mu$m membrane, the generated oligosaccharides were analyzed using matrix-assisted laser desorption/ionization time of flight mass spectrometry (MALDI-TOF MS) and high-performance liquid chromatography (HPLC) with protocols described previously (*Zhang et al., 2015*; *Zhang et al., 2020*). Since the content of produced DP4$_{ox}$ is a feasible parameter to evaluate the efficiency of LPMO reactions (*Zhang et al., 2020*), the peak areas of DP4$_{ox}$, DP5$_{ox}$, and DP6$_{ox}$ were calculated for comparative analysis.

## Synergetic assays

The chitinase synergetic experiments were carried out in 20 mM PBS buffer (pH 6.0) containing 5 mg mL$^{-1}$ $\alpha$-chitin (Sigma, USA), 1 $\mu$M $Sm$ChiB (anexo-type chitinase from

*Serratia marcescens*), 1 µM LPMO (*Bt*LPMO10A or *Bt*LPMO10B) and 1 mM ascorbic acid mixed in 500 µL reaction. The mixtures were incubated at 30 °C for 4, 8, 12, 24, 48, 72 h with an 800 rpm shaking. After separated the products from the reaction mixture by filtration through a 0.22 µm membrane, an equal volume of acetonitrile was added into the product solution. For *Bt*LPMO10A and *Bt*LPMO10B synergy studies, 0.5 µM *Bt*LPMO10A and *Bt*LPMO10B was added into the mixture. The reaction without the presence of LPMO was used as the control. The $(GlcNAc)_2$ released from the reactions was analyzed by HPLC equipped with an X-Amide column and a UV detector at 195 nm. The concentration of $(GlcNAc)_2$ in samples was calculated using commercial $(GlcNAc)_2$ as a standard. All experiments were performed in triplicates.

# RESULTS

## Structure and sequence analysis of *Bt*LPMO10A and *Bt*LPMO10B

*Bt*LPMO10A and *Bt*LPMO10B shares a sequence identity of 61%, and both enzymes contain a typical active site of AA10 family LPMOs that are comprised of a type II copper ion coordinated with two fully conserved histidines (Figs. 1A and 1B). A Phenylalanine residue was identified to be the axial residue of the copper ion. Furthermore, the residues in the substrate binding surface, as shown in Fig. 1B, were also highly conserved, which may involve in substrate binding similar as described in *Sm* CBP21A (*Vaaje-Kolstad et al., 2005*). Structure-based sequence alignment indicated that *Bt*LPMO10A and *Bt*LPMO10B possess similar loop regions surrounding the catalytic center (Fig. 1C).

## Substrate binding ability and activity of *Bt*LPMO10A and *Bt*LPMO10B

The substrate-binding ability of *Bt*LPMO10A and *Bt*LPMO10B on $\alpha$-, $\beta$-chitin, and colloidal chitin were assessed. As shown in Fig. 2B, *Bt*LPMO10A exhibits comparable binding ability to all three types of chitins. In contrast, for *Bt*LPMO10B, 80% of the enzyme protein was found bound to the colloidal chitin, which is significantly higher than the percentages of protein bound to the crystalline $\alpha$- and $\beta$-chitin. Deletion of the extra domains of *Bt*LPMO10B significantly decreased its binding ability towards all three kinds of chitins tested (Fig. 2B).

## Product analysis of *Bt*LPMO10A, *Bt*LPMO10B, and *Bt*LPMO10B-CD

The enzymatic activity of *Bt*LPMO10B towards various types of chitins, including $\alpha$-, $\beta$-chitin, colloidal chitin, and chitin oligosaccharides were investigated using MALDI-TOF MS. As shown in Figs. 3A and 3B, *Bt*LPMO10B and *Bt*LPMO10B-CD can act on $\alpha$-, $\beta$-chitin, and colloidal chitin and generate a product profile with even-numbered oxidized oligosaccharides as the dominant products. The products of *Bt*LPMO10A, *Bt*LPMO10B and *Bt*LPMO10B-CD from different kinds of chitins were further analyzed by HPLC (Table 1, Fig. 3D). The results showed that the $DP4_{ox}$ and $DP6_{ox}$ were the main oxidation products for *Bt*LPMO10A alone or in combination with *Bt*LPMO10B on almost all tested chitins. Differently, the $DP4_{ox}$ was the predominant product for *Bt*LPMO10B reactions and the highest production of the $DP4_{ox}$ was obtained when $\beta$-chitin was used as the substrate. The deletion of the CBM5 from *Bt*LPMO10B significantly reduced the production
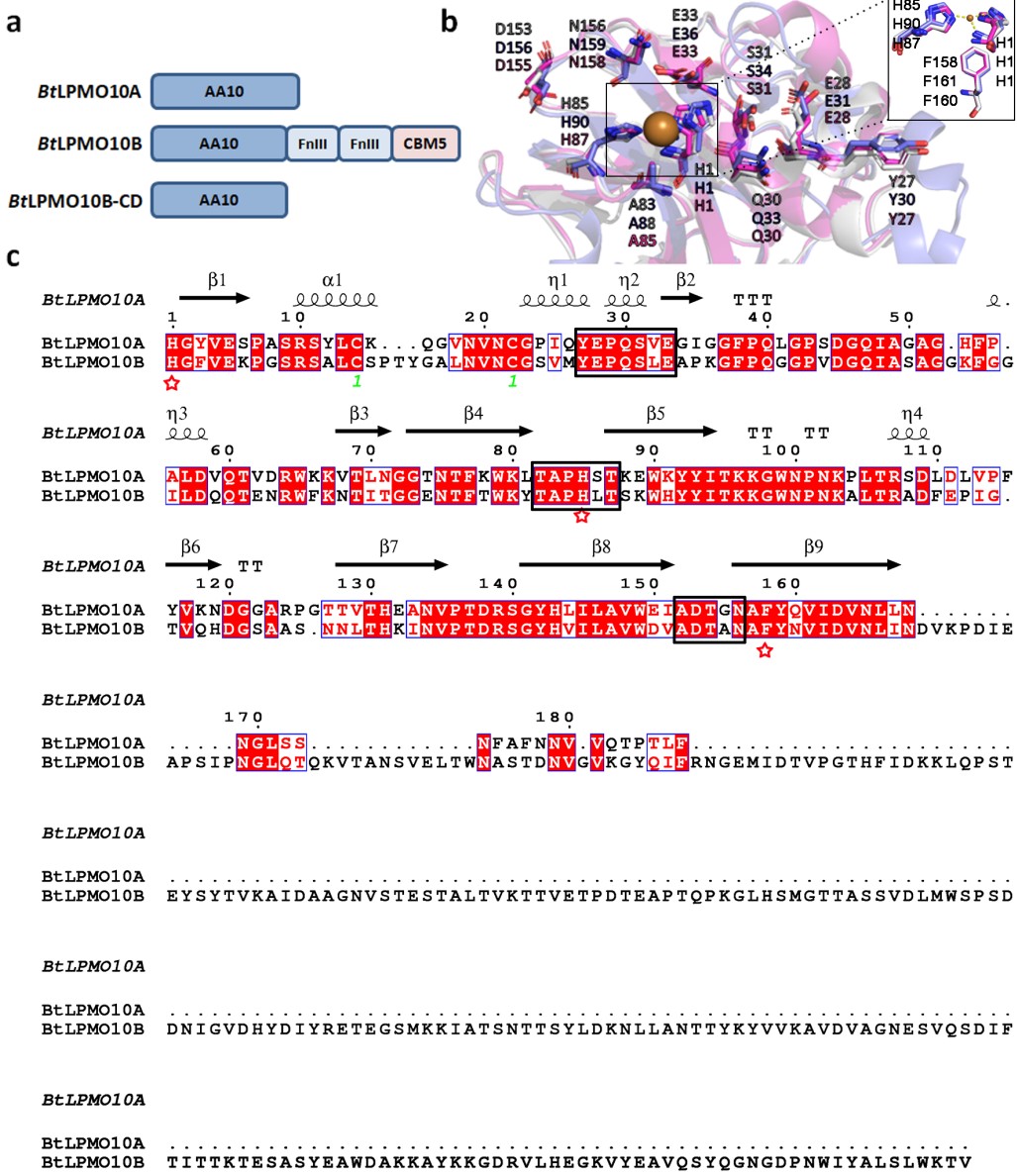

**Figure 1** **Structure and sequence comparison of _Bt_ LPMO10A and _Bt_ LPMO10B.** (A) Domain structure of _Bt_ LPMO10A, _Bt_ LPMO10B and _Bt_ LPMO10B-CD(1 to 178 amino acids in _Bt_ LPMO10B). (B) Close-up view of the substrate-binding surfaces of _Sm_ CBP21A (purple), _Bt_ LPMO10A (gray) and _Bt_ LPMO10B-CD (blue) which was modeled using _Ba_ LPMO10A (PDB ID: 2YOX) as template. The residues putative interacting with chitin is shown. (C) Structure-based sequence alignment of _Bt_ LPMO10A and _Bt_ LPMO10B. The amino acids from different loop regions surrounding the catalytic center were highlighted in the black rectangle and the residuescoordinated with the copper ion are marked by red stars.

of DP4$_{ox}$, while relative mild reduction of observed for DP5$_{ox}$ and DP6$_{ox}$ (Table 1). Moreover, although the binding ability of _Bt_ LPMO10B-CD toward colloidal chitin was significantly lower than the full-length enzyme (Fig. 2B), the amount of oxidized products (DP4$_{ox}$, DP5$_{ox}$, and DP6$_{ox}$) generated by _Bt_ LPMO10B-CD were similar as compared to

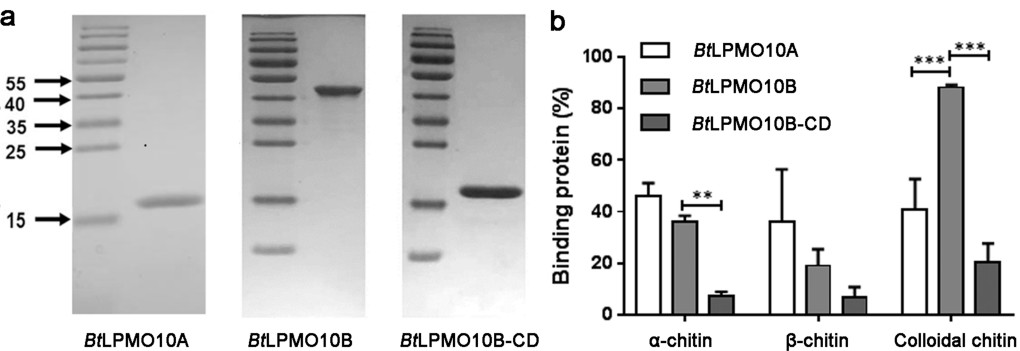

**Figure 2 Substrate binding ability and activity of *Bt*LPMO10A, *Bt*LPMO10B and *Bt*LPMO10B-N.** (A) Determination of the purity of *Bt*LPMO10B and *Bt*LPMO10B-CD by SDS-PAGE. (B) Substrate binding ability of *Bt*LPMO10A,*Bt*LPMO10B, and *Bt*LPMO10B-CD towards different type of chitin. The proportion of the bound proteins was calculated based on the residual proteins in the supernatant.

**Table 1 Peak areas of $DP4_{ox}$, $DP5_{ox}$, and $DP6_{ox}$ in HPLC chromatogram.**

| Chitin forms | Enzyme combinations | Area ($\mu$V*s) | | |
|---|---|---|---|---|
| | | $DP4_{ox}$ | $DP5_{ox}$ | $DP6_{ox}$ |
| $\alpha$-Chitin | *Bt*LPMO10A+*Bt*LPMO10B | 2003029 | 820042 | 2231064 |
| | *Bt*LPMO10B-CD | 1202074 | 427881 | 1292101 |
| | *Bt*LPMO10B | 2145732 | 412200 | 1016363 |
| | *Bt*LPMO10A | 5064803 | 1872054 | 5236980 |
| | – | 0 | 0 | 0 |
| $\beta$- Chitin | *Bt*LPMO10A+*Bt*LPMO10B | 4229223 | 1652738 | 5399536 |
| | *Bt*LPMO10B-CD | 1592714 | 349987 | 1302021 |
| | *Bt*LPMO10B | 4776729 | 799804 | 1920114 |
| | *Bt*LPMO10A | 2805547 | 1382478 | 5238538 |
| | – | 0 | 0 | 0 |
| Colloidal chitin | *Bt*LPMO10A+*Bt*LPMO10B | 1301934 | 730362 | 1433976 |
| | *Bt*LPMO10B-CD | 1000352 | 478331 | 982513 |
| | *Bt*LPMO10B | 3558191 | 509818 | 1133726 |
| | *Bt*LPMO10A | 4929692 | 4228932 | 7464410 |
| | – | 0 | 0 | 0 |

**Notes.**
   -,  no enzyme was added.

*Bt*LPMO10B. The combination of *Bt*LPMO10A and *Bt*LPMO10B led to similar product profiles on both $\alpha$-chitin and colloidal chitin as compared to *Bt*LPMO10B. In contrast, the product profile on $\beta$-chitin with the combined *Bt*LPMO10A and *Bt*LPMO10B was similar to that of *Bt*LPMO10A. These results suggest that *Bt*LPMO10B can affect the activity of *Bt*LPMO10A on $\alpha$-chitin and colloidal chitin.

## $H_2O_2$ production of *Bt*LPMO10A, *Bt*LPMO10B and *Bt*LPMO10B-CD

When the substrate was absent, $H_2O_2$ produced by *Bt*LPMO10A was much higher than that produced by *Bt*LPMO10B and *Bt*LPMO10B-CD (Figs. 4A and 4B). Accordingly, a much

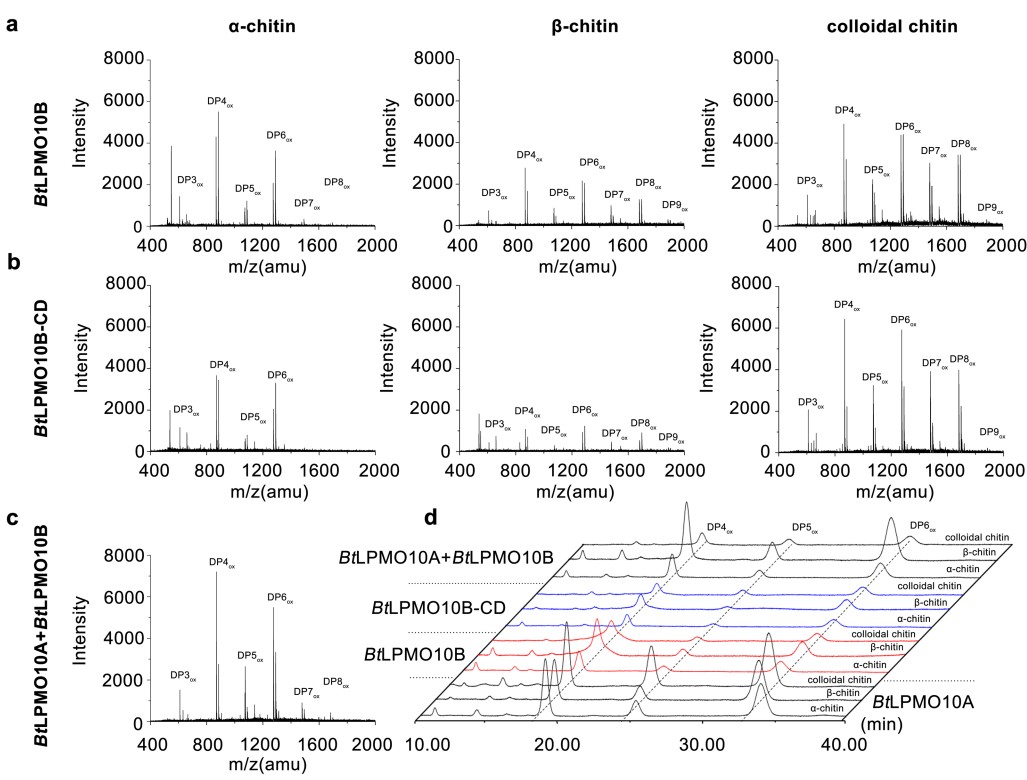

**Figure 3  Product analysis of *Bt*LPMO10A, *Bt*LPMO10B, and *Bt*LPMO10B-CD.** The products of LP-MOs towards $\alpha$-chitin, $\beta$-chitin and Colloidal chitin analyzed by MALDI-TOF MS (A, B, and C) or HPLC (D). The reactions were performed using 1 μM of enzymes and 1 mM of ascorbic acid.

stronger suppression of $H_2O_2$ generation in *Bt*LPMO10A was observed as the substrate, especially $\alpha$-chitin or colloidal chitin, has been provided. Similarly, a mild suppression of $H_2O_2$ production by substrate had been observed in the *Bt*LPMO10B, in which all three substrates showed comparable effect. As for the *Bt*LPMO10B-CD, similar inhibition in $H_2O_2$ production could be observed when colloidal chitin has been added, while negligible effect on the $H_2O_2$ production was recorded when provided with $\alpha$-chitin or $\beta$-chitin.

## The synergy in chitin degradation

The synergetic effects of *Bt*LPMO10A or *Bt*LPMO10B with *Sm*ChiB in $\alpha$-chitin degradation were performed in the present study (Fig. 5). *Sm*ChiB from *Serratia marcescens* is a model GH18 exo-chitinase which degrades the polymer chains from their non-reducing ends and dominantly produces $(GlcNAc)_2$ (*Chen et al., 2020*; *Van Aalten et al., 2000*). The synergy experiment results showed that when only *Sm*ChiB was present, the concentration of generated chitobiose reached its plateau ($0.336 \pm 0.0187$ mg/ml) after 12 h of reaction. In contrast, the additional supply of *Bt*LPMO10A or *Bt*LPMO10B can both significantly boost the accumulation of chitobiose, which has reached 1.464 mg ml$^{-1}$ for *Bt*LPMO10A and 1.232 mg ml$^{-1}$ for *Bt*LPMO10B, respectively, after 72 h of incubation. Moreover, when both *Bt*LPMO10A and *Bt*LPMO10B were provided, the curve of the concentration of $GlcNAc_2$ over time was similar to that observed when only *Bt*LPMO10B was provided.

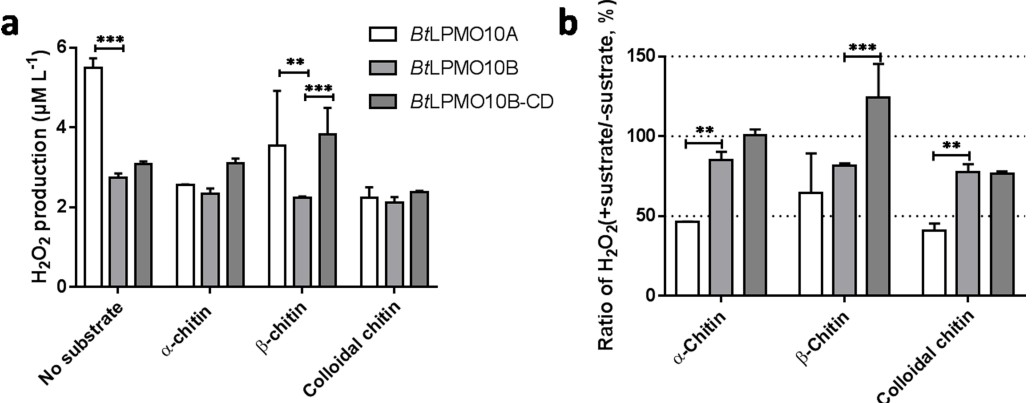

**Figure 4** **H₂O₂ production of *Bt*LPMO10A, *Bt*LPMO10B and *Bt*LPMO10B-CD.** (A) H₂O₂ production of the enzymes in the absence or presence of diverse substrates. (B) The ratio of H₂O₂ accumulation in the presence of substrates compared with the absence of substrates.

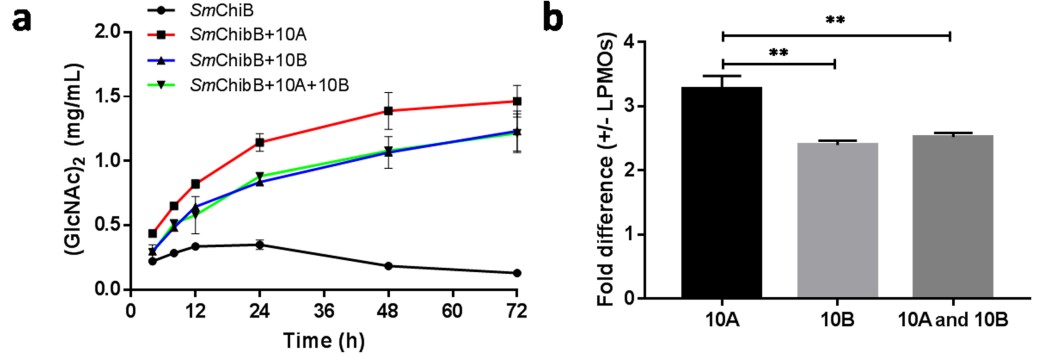

**Figure 5** **Synergetic effects of *Bt*LPMO10A (10A) and *Bt*LPMO10B (10B) to chitinase.** (A) Time course of (GlcNAc)₂ production in synergistic or non-synergistic reactions. (B) Fold difference of the production of (GlcNAc)₂ at 24 h between the synergistic and non-synergistic reactions.

## DISCUSSION

The present work was carried out to investigate the biochemical characteristics of *Bt*LPMO10A and *Bt*LPMO10B. It is well known that both *Bt*LPMO10A and *Bt*LPMO10B have a typical AA10 catalytic domain. Sequence and structure analysis indicated that both enzymes have a typical carbohydrate-binding surface and a distinct active site with a phenylalanine rather than tyrosine as the axial residual (*Forsberg et al., 2014*; *Span & Marletta, 2015*; *Vaaje-Kolstad et al., 2017*). The residues in the catalytic domains of both enzymes that may participate in the substrate binding are highly conserved as compared to those found in *Sm*CBP21A (*Vaaje-Kolstad et al., 2005*) (Fig. 1B). Substrate binding assays showed that *Bt*LPM10A has a similar binding ability to all three chitins tested (α-chitin, β-chitin, and colloidal chitin). In contrast, the multi-modular *Bt*LPMO10B is more inclined to bind to the colloidal chitin. As for the crystalline chitins, *Bt*LPMO10B

favors the $\beta$-form rather than the $\alpha$-form, which is contrary to the results reported by *Manjeet et al. (2019)* in which *Bt*LPMO10B (named *Bt*LPMO10A-FL) prefers to bind the $\alpha$-form chitin. Furthermore, *Bt*LPMO10B-CD only retained a small portion of the binding abilities of the full-length enzyme toward all three chitins, which are significantly lower than *Bt*LPMO10A. Interestingly, it exhibited comparable binding ability to both $\alpha$- and $\beta$-chitin which is different from the report by *Manjeet et al. (2019)* that *Bt*LPMO10B-CD showed no binding to $\alpha$-chitin while retained almost half the binding ability to $\beta$-chitin. This suggests that the substrate-binding capacity of *Bt*LPMO10B is mainly contributed by CBM5 which is in accordance to previous reports (*Manjeet et al., 2019*).

It is worth mention that the binding ability of *Bt*LPMO10B on colloidal chitin is not fully correlated with the activity indicated that certain amount of these binding is non-productive. To verify this possibility, $H_2O_2$ concentration in reaction mixtures with different chitins as the substrate were measured, which is based on the knowledge that a productive binding of LPMO to substrate will switch the enzyme from $H_2O_2$ production to consumption (*Wang, Walton & Rovira, 2019*; *Zhou et al., 2020*). As expected, the concentration of $H_2O_2$ showed only mild decrease in all three chitins. Different phenomenon was observed in *Bt*LPMO10A that the $H_2O_2$ concentration decreased significantly in the presence of the chitins, indicating its high binding efficiency. Moreover, when using colloidal chitin as the substrate, the $H_2O_2$ concentration in the reactions of *Bt*LPMO10B and *Bt*LPMO10B-CD were similar, despite their dramatic difference in binding ability to the substrate. These results indicated that the substrate binding of *Bt*LPMO10B enhanced by CBM5 is not led to enhanced substrate degradation. Therefore, the role of CBM in LPMOs may not just relate with enzyme catalytic activity.

To assess the potential application of *Bt*LPMO10A and *Bt*LPMO10B in chitin preparation, the synergetic effect of the two LPMOs with *Sm*ChiB was tested. The results showed that both enzymes can significantly improve the efficiency of the chitinase, similar as observed in other AA10 family LPMOs (*Forsberg et al., 2016*; *Mutahir et al., 2018*; *Nakagawa et al., 2015*; *Vaaje-Kolstad et al., 2012*). However, *Bt*LPMO10A exhibited much higher efficiency than *Bt*LPMO10B when synergized with *Sm*ChiB, which is consistent with the higher activity of *Bt*LPMO10A. Interestingly, when supplied with both *Bt*LPMO10A and *Bt*LPMO10B, the synergetic effect observed was similar to the condition that only supplied with *Bt*LPMO10B, which suggested that the contribution from *Bt*LPMO10A was almost fully suppressed by *Bt*LPMO10B. This may due to the higher binding efficiency of *Bt*LPMO10B on $\alpha$-chitin which hampered the binding of *Bt*LPMO10A.

## CONCLUSIONS

In summary, by comparing the structural and biochemical characteristics of *Bt*LPMO10A and *Bt*LPMO10B, we discovered that the two enzymes with highly conserved catalytic domains exhibit different substrate preferences. Further studies indicated that the C-terminal CBM5 domain of *Bt*LPMO10B may be responsible for these diversities implying that the two enzymes may function at different stages in the chitin degradation process.

These findings will help us better understand the biological reasons for the LPMO diversity and develop more efficient polysaccharide degrading enzyme cocktails.

### Funding

This work was supported by the ANSO Collaborative Research Program (ANSO-CR-KP-2020-14), the Chinese National Nature Science Foundation (31971217), the Dalian Science and Technology Innovation Fund-Key&Major Subject (2020JJ25CY017). Heng Yin was supported by the Outstanding Member Fund of CAS Youth Innovation Promotion Association (Y201939). Tang Li was supported by the Doctoral Scientific Research Foundation of Liaoning Province (2020-BS-013). The funders had no role in study design, data collection and analysis, decision to publish, or preparation of the manuscript.

### Grant Disclosures

The following grant information was disclosed by the authors:
ANSO Collaborative Research Program: ANSO-CR-KP-2020-14.
Chinese National Nature Science Foundation: 31971217.
Dalian Science and Technology Innovation Fund-Key&Major Subject: 2020JJ25CY017.
Outstanding Member Fund of CAS Youth Innovation Promotion Association: Y201939.
Doctoral Scientific Research Foundation of Liaoning Province: 2020-BS-013.

### Competing Interests

Heng Yin is an Academic Editor for PeerJ.

### Author Contributions

- Huiyan Zhang conceived and designed the experiments, authored or reviewed drafts of the article, and approved the final draft.
- Haichuan Zhou performed the experiments, prepared figures and/or tables, authored or reviewed drafts of the article, and approved the final draft.
- Yong Zhao conceived and designed the experiments, authored or reviewed drafts of the article, and approved the final draft.
- Tang Li analyzed the data, prepared figures and/or tables, authored or reviewed drafts of the article, and approved the final draft.
- Heng Yin performed the experiments, analyzed the data, prepared figures and/or tables, authored or reviewed drafts of the article, and approved the final draft.

### Data Availability

   The raw measurements and calculations for Figures 2, 4, and 5 are available in the Supplementary Files.

### Supplemental Information

Supplemental information for this article can be found online at http://dx.doi.org/10.7717/peerj.14670#supplemental-information.

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
