# Peer review of "Comparative studies of two AA10 family lytic polysaccharide monooxygenases from Bacillus thuringiensis"

_PeerJ, doi:10.7717/peerj.14670_

## Round 0.1 · original submission · Major Revisions

Please consider all the reviewer's questions, and present a detailed rebuttal letter and an amended version.

To expedite acceptance, the English language must meet the Journal criteria, before the article can be Accepted.

Reviewer 1 ·

Basic reporting

The paper has minor grammatical errors and typos. For example, space before comma, absense of oxford comma, Line 147 needs to be rephrased, etc.
The literature reference are sufficient and enough background is provided.

Experimental design

The choice of the template used in Modeller is questionable. The authors point that BtLPMO10A and BtLPMO10B share 61% sequence similarity and BtLPMO10A's crystal structure is available in PDB. Why was another template used? What was the sequence similarity of BtLPMO10B and BaLPMO10A?

Lines 225-226 mention the axial residue phenylalanine, but Fig1b does not show this residue. Needs to be incorporated.

Validity of the findings

The results validate the hypothesis and conclusions are well stated.

Additional comments

None

Reviewer 2 ·

Basic reporting

The manuscript is welcome, due to is important to know the possible application of polysaccharide monooxygenases, additionally Bacillus thuringiensis is a source of enzymes that must be studied to use the second most abundant biopolymer on Earth. Its is clear, figures are relevant and raw data are supplied.

Experimental design

The experimental design is well defined.

Validity of the findings

Conclusion are stated.

Additional comments

The manuscript is welcome, due to is important to know the possible application of polysaccharide monooxygenases, additionally Bacillus thuringiensis is a source of enzymes that must be studied to use the second most abundant biopolymer on Earth.

IIn this sense the authors must be cleared some issues:

According to the peak areas of table 1, the most important products are DP4ox, DP5ox and DP6ox. Do these products have have a particular application in the food or medicinal industry, and are these products involved in nature in some mechanism of sensing or are just serve as a carbon source for degrading microorganisms?


-In synergistic assays the authors mentioned the use of the well-known properties of SmChiB, but it will be helpful for the readers if the authors pointed out some of the SmChiB properties.

-In Enzymatic reactions and synergistic assays, the authors conducted their experiments with an agitation of shaking speed of 800 rpm, such agitation speed will be a condition in industrial application? Although, this manuscript establishes the basis of reaction conditions, how can we overcome this issue in the development of more efficient polysaccharide degrading enzyme cocktails.

---

## Round 0.2 · Minor Revisions

Please consider the comments by the reviewers and submit an amended version.

Reviewer 1 ·

Basic reporting

None

Experimental design

None

Validity of the findings

Fig 1c is a new concern for this reviewer.
The label in the new inset shows H161. It is not histidine but Phenylalanine. So, F161. Also, the residue numbering is not the same as in the alignment shown below Fig 1c. Fig 1b, is a superposition of three structures and authors have indicated all three residues for the ones highlighted in stick representation. In which case, which structure's residue numbering is followed in the inset? I am presuming it is BtLPMO10B. All three residue numbers from all three sequences should be listed for clarity.

Additional comments

None

Reviewer 2 ·

Basic reporting

The authors made appropriate clarifications.

Experimental design

No comment

Validity of the findings

No comment

Additional comments

No comment

---

## Round 0.3 · accepted · Accept

Thanks for addressing the comments, your paper is now accepted.